# Customer Perception and Its Influence on the Financial Performance in the Ecuadorian Banking Environment

Ana Belen Tulcanaza-Prieto [1,*] , Iliana E. Aguilar-Rodríguez [1] and Chang Won Lee [2]

1 Escuela de Negocios, Universidad de Las Américas, UDLA, Vía a Nayón, Quito 170124, Ecuador; iliana.aguilar@udla.edu.ec
2 School of Business, Hanyang University, Seoul 04763, Korea; leecw@hanyang.ac.kr
* Correspondence: ana.tulcanaza@udla.edu.ec

**Abstract:** This study examines the relationship between customer perception and financial corporate performance in the Ecuadorian banking environment. A self-designed online questionnaire was carried out to gather information regarding customer perception factors (tangibility, trust and service guarantee, empathy, customer satisfaction, and customer loyalty), while the financial data of the Ecuadorian banks were attained from their annual financial indicator reports (financial efficiency and liquidity). A total of 243 questionnaires were recollected and 219 were considered as the final valid data. SPSS 26 was utilized for data analyses. Structural equation modeling was used to test the hypotheses. Our findings revealed that customer perception has a positive and significant (at least at the 5% level) impact on the financial performance of banks. Similarly, customer loyalty is influenced by tangibility, trust and service guarantee, empathy, and customer satisfaction. Study results are mostly consistent with the banking environment of other countries, especially Nigeria, and Scandinavian nations. Bank managers might always prioritize the customer perception of the bank due to it being considered a strong predictor of the bank's financial performance. This study provides a complete statistical and econometrical model with tangible and intangible factors of customer perception (qualitative variables) and it includes financial records (quantitative variables). The main limitation of our study is the calculation of the customer perception index because Ecuadorian institutions do not calculate it and thus, we estimate the index in our model. For future research, the suggestion is that a corporate governance index with a customer perception component is included to improve the model.

**Keywords:** customer perception; financial corporate performance; Ecuadorian banking environment

## 1. Introduction

Customers are considered one of the most important stakeholders in any firm because companies are not likely to succeed without them. Kotler and Armstrong [1] mentioned that customer satisfaction is a vital factor in customers' perceptions of bank services, and thus, customer perception will be calculated based on how satisfied clients are with the quality of the bank services offered. Perception might include thoughts and impressions about the bank. In the actual competitive business environment, to sustain the firms' growth and raise their participation in the market, as well as increase their financial performance, banks need to understand and measure how to satisfy their customers, which also plays a crucial role in establishing a long-term bank–client relationship [2].

Furthermore, it is known that customer perceptions positively affect financial performance [3]. People purchase financial services with different benefits, resulting in diverse levels of behavior and satisfaction essential in reinforcing trust, commitment, and purchase intentions [4]. Therefore, satisfied customers continue to carry out business with the company and could become loyal, generating lower costs for the firm [5].

Competitive companies require knowledge that their primary role is to establish long-term customer relationships [6]; they must understate the gap between customers'

expectations and perceived service. Hence, investing in customer relationships allows for developing strategies that create value, thereby generating sustainable competitive advantage, bringing solid financial performance for the firms. However, evidence is still needed to know what factors of customer perception influence the emerging companies' financial performance.

Moreover, although satisfaction, loyalty, and financial performance affect customer experience [7], it is not yet known how the customer perception of financial services carries weight on financial performance. Additionally, the customer experience is contingent on the influence exerted by other people; hence, improving this experience can be a way to increase monetary returns [8].

In this context, the purpose of this study is to identify the relationship between customer perception and corporate performance in the Ecuadorian banking industry, employing five criteria for customer perception: tangibility, trust and service guarantee, empathy, customer satisfaction, and customer loyalty, while financial performance is measured by financial indicators: financial efficiency and liquidity. This study provides significant insights for bank managers and researchers. Loyal customers are considered a key developer of the bank's financial position because they identify their perspective and wishes with the bank projection, thus, customer opinion positively influences an entity's performance. On the other hand, managerial plans might strategically allocate the bank's financial resources to customer perception factors and investments in service quality [9] because managers can improve the customer perception by including customers' ideas, desires, and needs in all of a firm's processes. This study is in a position to provide managers with a complete view of which factors influence customer perception and the extent to which they affect the financial performance of banks.

The research proposes a new empirical model of the relationship between customer perception factors and the financial performance in the Ecuadorian banking context using two innovative metrics, financial efficiency and liquidity, as dependent variables. The study findings revealed a significant positive relationship between all customer perception factors and firm financial performance; therefore, the study results can be used as a guideline by managers to improve the customer perception factors in their organizations. If managers want to enhance customers' loyalty, they need to reduce the gap between customer perception and customer expectation. This research may help banks understand how the customer perception factors (tangibility, trust and service guarantee, empathy, customer satisfaction, and customer loyalty) interact to influence the overall financial performance.

The study focuses on the banking market in Ecuador because banks provide undifferentiated financial products and services; thus, one of the best ways to discriminate and generate brand recognition is to provide a high quality level of customer attention. Moreover, the financial sector depends on maintaining a long-term relationship with its customers, given the nature of products and services. This study will provide strategic view for decision-makers to improve customer relations and corporate performance processes in a banking environment and similar settings.

## 2. Theory and Hypotheses

### 2.1. Customer Perception

The customer collects information about a product or service and interprets the information to create a meaningful image of the particular item. Customers are interested in the offered product and in all the additional elements of service that they receive [6]. For instance, a customer sees advertisements, promotions, reviews, social media feedback, etc. and then they develop an impression about the products they see. This process is called customer perception. The entire process of customer perception starts when a customer sees or obtains information about a particular product, and then the customer starts to build an opinion about the product. One of the best ways to increase the service level is by knowing the customers' perceptions, which is built by customer experiences and the satisfaction degree of the customer service and quality [7,8]. Therefore, the business's

success is settled by how strongly the store's image and its products and services meet the customer expectations [10].

Customer perception involves how customers feel about the products, services, and brand. It includes customer feelings related to the inspiration provoked by the firm and the present and future expectations of the business. Moreover, these inputs can help managers to identify the firm's opportunities and challenges and improve the firm's marketing plan and service delivery, which will be reflected in the growing business. Customer perception is influenced by the context, which includes how the buying decision is made and the interaction between a user and seller. Environmental or contextual items include physical, technical, personal, and social factors that determine customers' decision-making [11].

### 2.1.1. Determinants of Customer Perception
Tangibility

Tangibles are considered the aspects of service that the customer can feel without purchasing the service; they contain the visible aspects of service to improve the customer's perception, including equipment, staff, physical facilities, products, communication material, and appearance [12]. Moreover, tangibles include the visual images that help customers form impressions of the quality of service, which will have a positive effect on the customer's perception and the customer's contribution to profit [13]. Similarly, a study conducted in Kenyan banks indicated that 63.1% of the variation in customer perception and satisfaction is associated with tangibility [14]. Therefore, the first hypothesis is:

**Hypothesis 1 (H1).** *Tangibility will have a positive effect on customer perception.*

Trust and Service Guarantee

Trust is grounded on interpersonal and business interactions [15]. Trust is related to the perceived credibility and benevolence of the firm and its services. Specifically, credibility is associated with a customer's perception that the words and promises of a service firm can be trusted, while benevolence refers to a customer's belief that a service provided by a firm is beneficial to its customers [16]. Moreover, trust allows customers to share personal information based on a belief that the information stays confidential; thus, if the service provider is considered trustworthy by customers, there is a higher possibility that the customer–firm relationship continues to grow and develop [17], which denotes that trust is a fundamental key in customer perception. On the other hand, a service guarantee might improve the competitive advantage of firms and allow customers to obtain high-quality products and services, which also increases the firm's trust and improves the customer perception of firm services. A superior service guarantee is a key tool for achieving and maintaining higher service quality and might be considered a determinant of a firm's success [18]. Therefore, the second hypothesis is:

**Hypothesis 2 (H2).** *Trust and service guarantee will have a positive effect on customer perception.*

Empathy

Empathy combines the interaction and communication between employees and customers grounded on altruistic motivation and pro-social behavior [19]. Empathy includes cognitive and emotional dimensions. Cognitive aspects involve service employees' capability in interpreting the customer's view by understanding their mind, thoughts, and intentions [20], while the emotional perspective induces employees' skills to help customers, including interpersonal and emotional concerns [21]. For better customer perception, employees must recognize and deal with customer needs using empathy as an important tool to anticipate the customer's thoughts and beliefs. Therefore, the third hypothesis is:

**Hypothesis 3 (H3).** *Empathy will have a positive effect on customer perception.*

Customer Satisfaction

Customer satisfaction is defined as the customer's emotional reaction to the perceived difference between performance and expectation of the product or service, which is also related to the long-term customer behavior and the customer's purchase intention [22]. Customer satisfaction is a key factor in the achievement of goals in the service environment because it evaluates the customer's behavior after the purchase of tangible and intangible products, and determines the satisfaction level of the customer [23]. Customer satisfaction includes cognitive and affective determinants. The cognitive step involves confirmations and expectations, while affective factors comprise inequity, performance, and realization [24]. Prior studies have shown a positive relationship between customer satisfaction and customer perception, and both dimensions positively on the firm performance [25,26]. Customers' satisfaction might make them buy more products and promote products to other people through word of mouth, which increases the firm's possibility of achieving a profit. Therefore, the fourth hypothesis is:

**Hypothesis 4 (H4).** *Customer satisfaction will have a positive effect on customer perception.*

Customer Loyalty

Customer loyalty comprises important attributes that satisfy customer needs and establish a long-term relationship between the firm and customers [24]. Loyalty refers to the steps of buying repeatedly, which also reflects positively on the company's profit. Customer loyalty can be analyzed by attitude and emotional characteristics. Behavioral loyalty refers to the occurrence of buying from a specific retailer, while emotional loyalty mentions the customer's concern based on previous experience and attitudes [27]. Therefore, if customers are not satisfied, they will have different options to claim, and they will move to other competition to cover their needs. Prior studies have revealed that factors influencing customer loyalty are tangibility, trust and service guarantee, empathy or caring, and customer satisfaction [28]. Using the theoretical background and findings of previous studies, the following fifth set of hypotheses are:

**Hypothesis 5 (H5).** *Customer loyalty will have a positive effect on customer perception.*

**Hypothesis 5a (H5a).** *Tangibility will have a positive effect on customer loyalty.*

**Hypothesis 5b (H5b).** *Trust and service guarantee will have a positive effect on customer loyalty.*

**Hypothesis 5c (H5c).** *Empathy will have a positive effect on customer loyalty.*

**Hypothesis 5d (H5d).** *Customer satisfaction will have a positive effect on customer loyalty.*

Thus, the sixth hypothesis refers to the multiple effects of all determinants on customer perception:

**Hypothesis 6 (H6).** *Customer perception is a second-order multidimensional construct comprising five dimensions: tangibility, trust and service guarantee, empathy, customer satisfaction, and customer loyalty.*

*2.2. Financial Performance*

Financial performance is a complete evaluation of a firm's overall performance using financial data expressed in monetary units, including, assets, liabilities, equity, expenses, revenue, and profitability. For internal users, financial performance is examined to determine their respective firms' well-being and standing, among other benchmarks. For external users, financial performance is analyzed to determine potential investment opportunities.

In this study, we focus on the impact of customer perception on the financial performance of the Ecuadorian banking industry, employing financial indicators such as financial efficiency and liquidity. Financial efficiency includes a series of strategies and mechanisms that produce enhanced conservation results compared to costs obtained by operational, fiscal, or social mechanisms [29]. On the other hand, liquidity refers to the facility of an asset, or security, to convert into cash without affecting its market price [30]. Both financial ratios (efficiency and liquidity) allow for analyzing how proficiently assets and managed liabilities are used in the short and long term, showing the level of the financial health of a firm.

*2.3. Customer Perception and Financial Performance in the Banking Industry*

Customers are the business's key assets, and thus, customer perception can be considered a primary goal of any firm. Perception involves everything in business because the customers perceive the firm impacts acquisition and retention, which will affect the ability to raise capital, showing that firm performance success depends on creating a positive customer perception. Satisfied customers continue their dealings with the firm, provoking less cost for the firm in maintaining these business relationships than acquiring a new customer [5]. Moreover, satisfied customers tend to demonstrate loyal behavior, which increases the firm's financial outcomes. Though customer perception does not immediately impact the firm's financial performance, it can serve as a diagnostic tool and indicator for prospective growth [3]. In today's competitive business environment, banks need to understand that their primary role is to establish long-term customer relationships [6]. To build a strong relationship between banks and customers, banks need to minimize the gap between customers' expected service and customers' perceived service.

Empirical studies have shown a positive relationship between customer perception and financial performance in the banking industry, given the fact that banking products are largely undifferentiated, and therefore, customer perception of service becomes the major competitive advantage that impacts the financial position of the bank, which is market expansion [4,31,32]. Customer perception includes responding to customers' questions (time, manner, and media), the customer attention provided by bank employees, and the marketing mix for products and services. Reaching customer expectations is one way to distinguish a business by itself, and thus, the economic benefits for a firm can increase. Therefore, the seventh hypothesis is:

**Hypothesis 7 (H7).** *Customer perception will have a positive effect on the financial performance.*

**Hypothesis 7a (H7a).** *Customer perception will have a positive effect on the financial efficiency of the firm.*

**Hypothesis 7b (H7b).** *Customer perception will have a positive effect on the liquidity of the firm.*

### 3. Research Model

We analyze the effect of customer perception factors on the financial performance of the Ecuadorian banking industry. Figure 1 presents the research model for financial corporate performance.

We introduced two financial corporate performance metrics (Equation (1)). We used financial efficiency and liquidity as the dependent variable, while size is a control variable.

$$FP_{i,t} = \beta_0 + \beta_1\, CP_{i,t} + \beta_2\, Size_{i,t} + \varepsilon_{i,t}, \tag{1}$$

where $FP_{i,t}$ is the financial performance for bank $i$ in year $t$. It is composed of two metrics of financial efficiency and liquidity ($Liq_{i,t}$). Financial efficiency is calculated by financial efficiency over equity ($FE1_{i,t}$) and financial efficiency over assets ($FE2_{i,t}$). $FE1_{i,t} = \left( \frac{Intermediation\ margin}{Average\ equity} \right)_{i,t}$ for bank $i$ in year $t$, $FE2_{i,t} = \left( \frac{Intermediation\ margin}{Average\ assets} \right)_{i,t}$ for

bank $i$ in year $t$, $Liq_{i,t} = \left(\frac{Current\ assets}{Current\ liabilities}\right)_{i,t}$ is the liquidity for bank $i$ in year $t$, $CP_{i,t}$ is the customer perception for bank $i$ in year $t$, $Size_{i,t} = Log\ (Total\ assets)_{i,t}$ is the size for bank $i$ in year $t$ and is represented by natural logarithm of total assets, $\varepsilon_{i,t}$ is the error term for bank $i$ in year $t$.

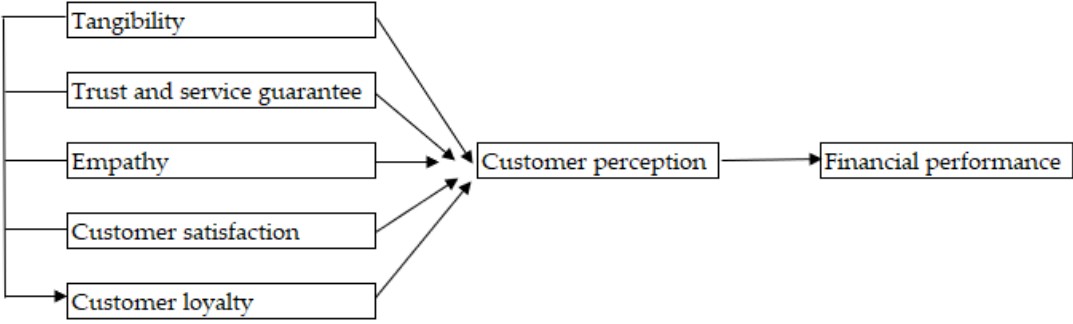

**Figure 1.** Research model of customer perception determinants and financial performance.

Previous studies have included profitability measures as the common financial performance metrics. This manuscript innovated the financial performance metrics using financial efficiency and liquidity. The purpose of the study using these metrics is to identify the insights about the business and how to potentially improve the health of the organization. Moreover, efficiency ratios determine how productively a firm manages its intermediation margin, assets, and liabilities to maximize profits. Shareholders are interested in financial efficiency ratios to assess how effectively their investments in the firm are being used. On the other hand, liquidity ratio determines the firm's ability to cover its short-term debt obligations. Healthy liquidity will help managers to overcome a firm's financial challenges, secure loans, and plan their financial future.

*Measurement of Constructs*

An online survey was designed to investigate the effect of customer perception on the financial performance in the Ecuadorian banking industry. The study consists of five constructs: tangibility, trust and service guarantee, empathy, customer satisfaction, and customer loyalty. The determinants of customer perception were measured by multiple items. Each item employs a five-point Likert scale: 1 represents a strongly disagree opinion, and 5 signifies a strongly agree opinion. The study adapted the items from prior research to ensure content validity. It is important to mention that most Ecuadorian private enterprises, including banks, are family-owned businesses. The conservative owners of these businesses would not easily reveal their actual financial data, especially if these data contain sensitive and classified information [9]. Thus, this study selected and designed special items for each construct to achieve the research objective. The final list of items for each construct is presented in Table 1.

**Table 1.** Scale items for constructs.

| Constructs | Items | Label | Related Literature |
|---|---|---|---|
| Demographic information | Gender, age, marital status, level of education, occupation, monthly income, banking entity | Nominal scale | |
| Tangibility (Tan) | My bank owns visually attractive facilities. | Tan1 | [24,33,34] |
| | My bank has modern-looking technology equipment. | Tan2 | |
| | My bank has new physical and electronic security systems. | Tan3 | |
| | The staff of my bank uses their institutional uniform with a neat appearance. | Tan4 | |
| | My bank uses attractive and innovative materials (office supplies, cards, institutional branding). | Tan5 | |
| | I am satisfied with the monthly bank statement provided by my institution. | Tan6 | |

**Table 1.** *Cont.*

| Constructs | Items | Label | Related Literature |
|---|---|---|---|
| Trust and service guarantee (Tru) | I am satisfied with the physical and electronic security of my bank. | Tru1 | [35–37] |
| | I am satisfied with the knowledge and experience of the bank staff in the resolution of problems. | Tru2 | |
| | The staff of my bank demonstrates a sincere interest in solving my banking problems. | Tru3 | |
| | I am satisfied with the response time of my bank. | Tru4 | |
| | The staff of my bank shows confidence with me. | Tru5 | |
| Empathy (Emp) | The staff of my bank can establish the time that a certain banking or financial service will take. | Emp1 | [11,24,34,38] |
| | The staff of my bank provides prompt customer services using physical/digital/electronic tools. | Emp2 | |
| | The staff of my bank is always willing to help/guide the clients. | Emp3 | |
| | The staff of my bank offers personalized customer attention. | Emp4 | |
| | I am satisfied with the opening hours of my bank | Emp5 | |
| Customer satisfaction (Sat) | Overall, my feeling about this bank is satisfactory. | Sat1 | [33,39–42] |
| | My bank is one of the best three entities in the entire national financial system. | Sat2 | |
| | I would choose this bank again. | Sat3 | |
| | The staff of my bank demonstrates kindness in the customer's attention. | Sat4 | |
| | The staff of my bank has the knowledge and experience to answer my questions. | Sat5 | |
| Customer loyalty (Loy) | My bank fulfills its promises in the tome offered. | Loy1 | [11,33,39,40,43] |
| | I mention attributes of my bank. | Loy2 | |
| | I (will) recommend my bank to other customers. | Loy3 | |
| | I will continue to do more business with my bank in the future. | Loy4 | |
| | I am a loyal customer of my bank. | Loy5 | |

To reduce the endogenous relationship between variables, bias, and inconsistency in the parameter estimates, the study includes size as a control variable in the financial performance model [44]. Moreover, the dependent variable (financial efficiency and liquidity) error terms are not normally distributed, given that Pearson correlation coefficients between the main variables and the residuals for each model do not expose high and significant (at the 5% level) values [45].

## 4. Empirical Results

The online questionnaire was distributed and collected by Google Forms, while IBM SPSS Statistics 26 was employed to tabulate all the data. A total of 243 questionnaires were recollected. However, the final sample is formed by 219 observations due to the duplicated and incomplete responses.

### 4.1. Demographic Analysis

In the statistical characteristics of this study, 67.6% of men and 32.4% of women responded. Their percentage age is distributed as 36–45 years old at 44.7%, 26–35 years old at 35.6%, 46–55 years old at 9.1%, 18–25 and 56–65 years old at 4.6% in each category, and higher than 65 years old at 1.4%. The marital status of the respondents is disaggregated as married at 47.9%, single at 33.8%, free union at 11.9%, divorced or separated at 5.5%, and widower at 0.9%. Referring to academic life, masters and doctorate degrees occupy the highest percentage of respondents (43.4%), 29.2% are junior college graduates, 26.0% are college graduates, and 1.4% receive primary education. Private employees represent 45.7% of the total respondents, while 34.2% work in the public sector, 13.7% work by themselves (own job and entrepreneur), and the remaining percentage (6.4%) are housewives and students. The monthly income of the respondents is concentrated (51.1%) in the category of USD 450.00–USD 1200.00, 16.7% of respondents receive USD 1200.01–USD 1750.00 as

monthly salary, 13.7% of respondents receive more than USD 2500 per month, 13.2% obtain USD 1750.01–USD 2500.00 monthly, and 5.3% of respondents receive a monthly wage lower than 450.00. Most of those surveyed (79.9%) save and invest money in an Ecuadorian private bank and 20.1% have a bank account in an Ecuadorian public financial institution. Banco Pichincha, Banco Internacional, Banco del Pacífico are the three Ecuadorian banks preferred by the respondents (77.2%), the remaining percentage (22.8%) are distributed in other financial institutions.

### 4.2. Descriptive Statistics and Exploratory Factor Analysis

In this study, the rotation method was Oblimin with principal component analysis. The component correlation matrix revealed values higher than 0.3. The factor loading values were determined based on 0.5. Items Tan4, Tan6, Tru1, Emp5, Sat1, and Loy2 were removed and omitted in the subsequent analysis because they presented lower internal consistency and discriminant validity. Thus, the initial number of items was 26 and we reduced it to 20 items. Table 2 presents the descriptive statistics and exploratory factor analysis (EFA). In EFA, the Kaiser–Meyer–Olkim (KMO) was 0.958 (KMO > 0.5) while Bartlett's sphericity test significance was 0.000 (Sig. < 0.05).

**Table 2.** Descriptive statistics and exploratory factor analysis.

| Constructs | Label | Mean | Std. Deviation | Variance | Composite Mean | Factor Loadings |
|---|---|---|---|---|---|---|
| Tangibility (Tan) | Tan1 | 4.233 | 0.901 | 0.812 | 4.128 | 0.919 |
| | Tan2 | 4.187 | 0.942 | 0.887 | | 0.907 |
| | Tan3 | 4.037 | 1.057 | 1.118 | | 0.501 |
| | Tan5 | 4.055 | 0.942 | 0.887 | | 0.534 |
| Trust and service guarantee (Tru) | Tru2 | 3.758 | 1.185 | 1.404 | 3.752 | 0.805 |
| | Tru3 | 3.749 | 1.179 | 1.391 | | 0.906 |
| | Tru4 | 3.635 | 1.335 | 1.783 | | 0.790 |
| | Tru5 | 3.868 | 1.144 | 1.308 | | 0.709 |
| Empathy (Emp) | Emp1 | 3.795 | 1.087 | 1.182 | 3.862 | 0.679 |
| | Emp2 | 3.858 | 1.159 | 1.342 | | 0.900 |
| | Emp3 | 3.954 | 1.091 | 1.191 | | 0.920 |
| | Emp4 | 3.840 | 1.184 | 1.401 | | 0.965 |
| Customer satisfaction (Sat) | Sat2 | 4.201 | 1.107 | 1.226 | 4.078 | 0.946 |
| | Sat3 | 3.986 | 1.258 | 1.582 | | 0.725 |
| | Sat4 | 4.164 | 1.058 | 1.120 | | 0.588 |
| | Sat5 | 3.959 | 1.126 | 1.269 | | 0.831 |
| Customer loyalty (Loy) | Loy1 | 3.781 | 1.218 | 1.484 | 3.946 | 0.813 |
| | Loy3 | 3.804 | 1.246 | 1.553 | | 0.632 |
| | Loy4 | 4.219 | 0.980 | 0.961 | | 0.812 |
| | Loy5 | 3.982 | 1.211 | 1.468 | | 0.739 |

Note: N = 219. Kaiser–Meyer–Olkim (KMO) = 0.958. Significance of Bartlett's sphericity test = 0.000. Extraction sums of squared loadings (cumulative variance %) = 74.806%. Extraction method: principal component analysis. Rotation method: Oblimin. Factor extraction criteria: eigenvalue (1, 0).

On the five-point Likert scale, the composite score of Tan was 4.128, which was the highest value compared to the remaining customer perception factors ($\mu$ = 3.752–4.078). This result reflects that Ecuadorian banks focus on tangible components and their perception of customers; and thus, attractive facilities, technological equipment, electronic security systems, attractive and innovative materials are key components for banks and their customers.

The Sat composite score is 4.078, showing a high level of customer bank satisfaction. This finding is fundamentally aligned with the fact that customers believe that "their bank is one of the best three entities in the entire national financial system" ($\mu$ = 4.201) and they perceive the attention of the bank staff as kind ($\mu$ = 4.164). The respondents displayed a condensed sensitivity for loyalty ($\mu$ = 3.946), indicating that customers will continue

conducting business with their banks in the future. The composite score for Emp was 3.862, representing that the bank employees help and guide clients using physical, digital, and electronic tools ($\mu = 3.954$ and $\mu = 3.858$, respectively). The composite score for Tru was 3.752, which is the lower component in the perception for customers, shows that customers are confident with the bank staff given they solve customer's problems using their knowledge and experience ($\mu = 3.868$ and $\mu = 3.758$, respectively).

### 4.3. Reliability Analysis

Table 3 shows the descriptive statistics and correlation matrix for all constructs. Cronbach's alpha scored between 0.806 to 0.939, which was above the recommended level of 0.6 [46]. The composite reliability fluctuated from 0.820 to 0.927, which was higher than the suggested level of 0.7. The average variance extracted (AVE) values ranged from 0.551 to 0.762, which was greater than the proposed level of 0.5. There are no multicollinearity problems between variables because the correlation coefficients were lower than 0.7. The square root of AVE was larger than the correlation values between variables, showing adequate discriminant validity in the proposed model [47].

**Table 3.** Descriptive statistics and correlation matrix.

| Var. | Items | CA | CR | AVE | Correlations | | | | |
| | | | | | Tan | Tru | Emp | Sat | Loy |
|---|---|---|---|---|---|---|---|---|---|
| Tan | 4 | 0.806 | 0.820 | 0.551 | (0.742) | | | | |
| Tru | 4 | 0.939 | 0.880 | 0.649 | 0.603 *** | (0.806) | | | |
| Emp | 4 | 0.929 | 0.927 | 0.762 | 0.653 *** | 0.671 *** | (0.873) | | |
| Sat | 4 | 0.906 | 0.861 | 0.614 | 0.656 *** | 0.632 *** | 0.529 *** | (0.784) | |
| Loy | 4 | 0.873 | 0.838 | 0.566 | 0.660 *** | 0.636 *** | 0.520 *** | 0.586 *** | (0.753) |

Note: CA = Cronbach's alpha. CR = composite reliability. AVE = average variance extracted. Values in parenthesis are root AVE. *** indicates significance at the 1% level.

### 4.4. Regression Analysis

Table 4 presents the results of individual linear regressions to test the customer perception effect on financial performance in the Ecuadorian banking industry. The lowest adjusted R-Square is 0.433 (H5 result) and the highest value is 0.891 (H6 result). The main hypothesis of the study is supported, demonstrating that financial performance is positively affected by customer perception ($\beta = 0.007$, 0.001, and 0.009; $p < 0.05$, $p < 0.05$, $p < 0.01$, respectively). These results show the importance of satisfied customers, which generates a long-term business relationship between customers and banks [3,5]. Banks need to invest in marketing campaigns, which are focused on the improvement in client perceptions because banking products are similar between entities; thus, the only way to differentiate the banking service is through the customer sensitivities [31,32]. Moreover, loyal customers contribute to increasing the financial performance of banks because they might recommend their bank to others; thus, the number of clients and financial operations will grow for the bank. All findings are consistent with the customer-oriented perspective of banks and their efforts to build, maintain, and reinforce their reputation and customer relationship [9,48–50].

**Table 4.** Multiple regression results.

| Regression | Prop Effect | Adj. R$^2$ | Durbin Watson | F | Constant | β | Test Result |
|---|---|---|---|---|---|---|---|
| Tan → CP | + | 0.623 | 2.140 | 361.817 *** | 0.139 (0.681) | 0.924 *** (19.021) | H1 supported |
| Tru → CP | + | 0.888 | 1.978 | 1731.733 *** | 1.115 *** (15.675) | 0.756 *** (41.614) | H2 supported |

**Table 4.** *Cont.*

| Regression | Prop Effect | Adj. R$^2$ | Durbin Watson | F | Constant | β | Test Result |
|---|---|---|---|---|---|---|---|
| Emp → CP | + | 0.859 | 2.003 | 1330.745 *** | 0.834 *** (9.431) | 0.808 *** (36.476) | H3 supported |
| Sat → CP | + | 0.868 | 1.844 | 1431.415 *** | 0.575 *** (6.258) | 0.828 *** (37.834) | H4 supported |
| Loy → CP | + | 0.867 | 1.946 | 1420.837 *** | 0.648 *** (7.168) | 0.838 *** (37.694) | H5 supported |
| Tan → Loy | + | 0.433 | 2.091 | 167.380 *** | 0.406 (1.458) | 0.858 *** (12.938) | H5a supported |
| Tru → Loy | + | 0.698 | 1.889 | 505.077 *** | 1.146 *** (8.818) | 0.746 *** (22.474) | H5b supported |
| Emp → Loy | + | 0.671 | 2.174 | 445.024 *** | 0.879 *** (5.847) | 0.794 *** (21.096) | H5c supported |
| Sat → Loy | + | 0.794 | 1.892 | 791.940 *** | 0.375 *** (2.868) | 0.876 *** (28.142) | H5d supported |
| Tan ⎫ Tru ⎪ →CP Emp ⎪ Sat ⎭ | + + + + | 0.891 | 1.944 | 6020.643 *** | 0.036 (1.096) | 0.214 *** (19.913) 0.239 *** (20.077) 0.225 *** (18.224) 0.311 *** (27.685) | H6 supported |
| CP ⎫ →FE1 Size ⎭ | + Control var. | 0.443 | 1.914 | 19.196 *** | 0.689 *** (5.510) | 0.007 ** (1.981) −0.030 *** (−5.637) | H7a supported |
| CP ⎫ →FE2 Size ⎭ | + Control var. | 0.446 | 1.925 | 19.695 *** | 0.069 *** (5.457) | 0.001 ** (1.965) −0.003 *** (−5.626) | H7a supported |
| CP ⎫ →Liq Size ⎭ | + Control var. | 0.495 | 2.024 | 107.665 *** | 1.514 *** (15.592) | 0.009 *** (2.854) −0.057 *** (−13.847) | H7b supported |

Note: Beta corresponds to unstandardized coefficients. Numbers inside the parenthesis are t-statistics. *** and ** indicates statistical significance at the 1% and 5% level, respectively.

Tangibility, trust and service guarantee, empathy, customer satisfaction, and customer loyalty each presented a significant positive relationship (at the 1% level) with customer perception and are identified as key developers of the positive relationship between customers and bank performance in the Ecuadorian banking industry. The perception of Ecuadorian customers is that banks present attractive and modern facilities with a high level of technology and security, which is the support of H1 at the 1% level of significance. Moreover, hypotheses from 2 to 5 are accepted at the same level of confidence, showing knowledge and experience in the bank staff in solving customer problems; thus, the customer is identified with his/her entity, improving their bank loyalty.

Moreover, tangibility, trust and service guarantee, empathy, and customer satisfaction positively contribute to the increase in customer loyalty (H5a–H5d are supported at the 1% level). Customer loyalty results when the customer is convinced of and identifies with the bank's attitude and performance, which generates long-term corporate profitability and financial efficiency, grounded by the word-of-mouth recommendation, decreasing the marketing costs and customer retention [51,52].

## 5. Discussion

Customer perception of a bank suggests that factors such as tangibility, trust and service guarantee, empathy, customer satisfaction, and customer loyalty are key developers of the financial performance of the entity because customers evaluate a product or service in terms of whether that product or service has met their needs and expectations, and if that margin is short, customers will be loyal to the bank, showing an improvement in its financial indicators. The research results signaled how the robustness of customer perception was associated with the better financial development of the bank. Moreover, the research results show that tangibility, trust and service guarantee, empathy, and customer satisfaction influence customer loyalty positively and significantly (at the 1% level).

Customer perception of quality in banking services is an indefinable concept because of the intangible nature of the service provided by banks, which varies depending on the situation. Most authors on customer perception have mentioned that customer expectations are not necessarily predictable or consistent [6]. Aligned with this principle, customers of products and services tend to increase their quality standards of goods, raising the market competitivity; thus, there is a continuous increase in customer expectations and customer demands on the bank service quality [38]. As a result, banks are projected to meet the customer needs and demands, using effective and efficient marketing tools to retain them to increase the bank's financial position by the growth of high customer perception of the institution [1].

All items considered in this manuscript strongly affect the bank's financial performance. If the institution repeatedly satisfies a customer, this customer will continue conducting their transactions with the same bank [53]. Moreover, customer perception is associated with the customer image of their bank. Therefore, managers might always prioritize as a priority the customer perception of the institution, since it is considered a strong predictor of the bank's financial performance, which is accompanied by listening to customers' requirements and then creating products and services that satisfy them [33].

Ecuadorian banks and their internal policies to evaluate the customer perception factors need to acquire and assess correct data about customer needs, as better information leads to better products and services, which increase the customer satisfaction needs. More satisfied customers then buy the product or service again, thus ensuring the current and future firm performance. Moreover, managers need to consider the market competitors and need to introduce a benchmarking analysis to improve the quality of the banking service in order to maintain and increase their satisfied clients.

This study focused on the Ecuadorian banking sector of 24 private banks. This sector is known as a service or tertiary sector in the economy. According to the Ecuadorian Central Bank, the real Gross Domestic Product (GDP) of the Ecuadorian financial service activities represented 3.8% of Ecuador's real GDP in 2021, while the financial performance of the Ecuadorian private banks was 108.4 and 28.6 for financial efficiency and liquidity in December 2021. Furthermore, four Ecuadorian private banks (Banco Pichincha, Banco de Guayaquil, Produbanco, and Banco Bolivariano CA) concentrated 60.9% of the total assets of the top 10 Ecuadorian economic groups classified by their size during 2020. Ultimately, six Ecuadorian private banks are included in the top 10 Ecuadorian economic groups classified by their tax collection using the ranking 2016–2020. The total tax collection of these six banks represented 28.8% and 13.4% of the total tax collection of the Ecuadorian economic groups and the total national net tax collection during 2019, respectively [54].

## 6. Conclusions

We analyzed the relationship between customer perception and financial performance in the Ecuadorian banking industry. Using 219 records from a self-designed online questionnaire and financial efficiency and liquidity as a proxy for financial performance, we found that corporate financial performance is affected positively by customer perception at least at the 5% level. The positive relationship between customer perception and corporate financial performance is aligned with prior studies' findings in Nigerian, American, and

Scandinavian banks [3,6,32]. Furthermore, the increase in customer perception might be used as a primary business strategy to increase a bank's financial performance by improving the customer's attention standards and clients' satisfaction. Therefore, these activities are considered in the strategic and functional plan to raise goodwill and reduce promotional costs to customer loyalty.

Tangibility, trust and service guarantee, empathy, customer satisfaction, and customer loyalty are the tangible and intangible factors that composed customer perception, and the fourth first factors influenced positively and significantly on the fifth characteristic (customer loyalty) showing that loyal customers need and establish a long-term relationship with the firm [24,28] if they can satisfy their needs based on the previous, actual, and future experience and attitudes. Furthermore, banks provide undifferentiated products and services, and therefore, customer perception of service and the determinants of customer perception become the major competitive advantage between entities, which also impacts positively on the financial performance of the bank and its market position and expansion [31,32].

This paper contributes to prior literature by providing a complete statistical and econometrical model of tangible and intangible factors that compose customer perception, offering the possibility to measure and define the potential gaps between what customers expect and receive from their qualified banks. Second, this manuscript links customer perception (customer opinion, qualitative variables) with the financial records (quantitative variables) of the Ecuadorian banks, and therefore, the study showed a positive and significant relationship between both variables, which has not been deeply analyzed in the Ecuadorian context. Third, previous studies included profitability metrics as dependent variables. The study innovates the financial performance metrics introducing financial efficiency and liquidity as dependent variables, which involves firms' productivity, investment effectivity, firms' health, and firms' financial future plans.

The main limitation of our study is the direct access to customer perception data because Ecuador does not calculate a customer perception index for economic sectors; therefore, by the self-designed online questionnaire, we covered this need. The authors suggest focusing on the longitudinal analysis with financial and perception analyses for the Ecuadorian banking industry in future research. We also recommend including brand images and a corporate governance index with a customer perception component to improve this manuscript's statistical and econometric model.

**Author Contributions:** The authors contributed extensively to the work presented in this paper. Writing—original draft preparation, A.B.T.-P.; writing—review and editing, I.E.A.-R. and C.W.L. as a co-author. All authors have read and agreed to the published version of the manuscript.

**Funding:** We extend our gratitude and acknowledgment to the Universidad de Las Américas, which financially supported this research (EDN.ATP.22.01).

**Institutional Review Board Statement:** Not applicable.

**Informed Consent Statement:** Not applicable.

**Data Availability Statement:** The datasets used and analyzed in this study are available from the corresponding author on justified request.

**Conflicts of Interest:** The authors declare no conflict of interest. The funders had no role in the design of the study; in the collection, analyses, or interpretation of data; in the writing of the manuscript, or in the decision to publish the results.

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
