# Peer review of "Customer Perception and Its Influence on the Financial Performance in the Ecuadorian Banking Environment"

_sustainability, doi:10.3390/su14126960_

Round 1
Reviewer 1 Report
The manuscript "Customer Perception and Its influence on the Financial Performance in the Ecuadorian Banking Environment" examines "the relationship between customer perception and financial corporate performance in the Ecuadorian banking environment". For the most part, this manuscript is well written and the data that has been collected is interesting. However, I had several concerns pertaining to the theoretical contributions and the research design of this study. Below, I outline my concerns and my suggestions.
The topic addressed by the article is not particularly innovative, if we consider the purpose of understanding the correlation between external variables and the production of particular financial performances of a specific business sector. Specifically, the article focuses on Ecuador's banking sector and investigates its financial performance.
Undoubtedly, the issue of performance measurement system of business operators, more generally, represents an important activity for evaluating the performance of a company or a business sector, able to provide decision-makers the essential elements for an assessment of the choices made and to guide future choices and appropriate decisions to increase performance.
The performance measurement system must urge decision makers to undertake certain paths to increase the ability to survive in the contexts in which they operate. And this by investigating the cause-effect relationships between the observed performances and the factors on which they depend.
The article investigates the correlation between consumers 'perception of the Ecuadorian banking system and its financial performance, using the analysis of a series of factors considered by the authors to determine consumers' perception of the Ecuadorian banking system: Tangibility, Trust and service guarantee, Empathy, Customer satisfaction, Customer loyalty. Fig. 1 of Par. 3 attempts to explain the correlation between the aforementioned factors and the customers’ perception, therefore between the customers’ perception and financial performance, but it is believed that from a representative point of view, the figure should be revised in relation to the relationship between the aforementioned factors and consumers’ perception. In fact, the aforementioned factors that shape the perception of the consumer, therefore, the direction of the relationship between the factors and the customers’ perception is opposite to that represented in the figure.
Certainly, customer loyalty can be understood as a result of the previous factors, which can induce the customer to remain loyal to his bank.
The article intends to demonstrate the correlation between the consumer's perception of Ecuadorian banks - as resulting from the analysis of the aforementioned factors - and the financial performance of the Ecuadorian banking system; in particular considering its financial efficiency and the liquidity generated.
The research model adopted is interesting, recurring to the collection of data through 243 questionnaires proposed to the customers of the Ecuadorian banking system, in order to correlate - through linear regression analysis - the effects of customers’ perception and the financial performance of Ecuadorian banks.
Considering the research model adopted and the results obtained, an extension of the model could be suggested, in order to investigate other variables that cross with the factors considered in the construction of the customers’ perception, which could be able to influence the financial performance of the Ecuadorian banking system and, therefore, measure the level of influence.
Author Response
In the attached document, you can find our response.

Reviewer 2 Report
The manuscript titled “Customer Perception and Its influence on the Financial Performance in the Ecuadorian Banking Environment” aims to investigate the relationship between customer perception and financial corporate performances in the Ecuadorian banking environment. The study is interesting and already in high quality. Therefore, the study can be accepted in its current form
Author Response

(The authors gave the same response as above.)

Author Response

(The authors gave the same response as above.)

Round 2
Reviewer 1 Report
The authors addressed the main concerns from the reviews, the revised version of the manuscript appears to be good.